# Manifestations of metastable criticality in the long-range structure of model water glasses

Thomas E. Gartner III [1], Salvatore Torquato[1,2,3,4], Roberto Car[1,2,3,4] & Pablo G. Debenedetti [5✉]

Much attention has been devoted to water's metastable phase behavior, including poly-amorphism (multiple amorphous solid phases), and the hypothesized liquid-liquid transition and associated critical point. However, the possible relationship between these phenomena remains incompletely understood. Using molecular dynamics simulations of the realistic TIP4P/2005 model, we found a striking signature of the liquid-liquid critical point in the structure of water glasses, manifested as a pronounced increase in long-range density fluctuations at pressures proximate to the critical pressure. By contrast, these signatures were absent in glasses of two model systems that lack a critical point. We also characterized the departure from equilibrium upon vitrification via the non-equilibrium index; water-like systems exhibited a strong pressure dependence in this metric, whereas simple liquids did not. These results reflect a surprising relationship between the metastable equilibrium phenomenon of liquid-liquid criticality and the non-equilibrium structure of glassy water, with implications for our understanding of water phase behavior and glass physics. Our calculations suggest a possible experimental route to probing the existence of the liquid-liquid transition in water and other fluids.

[1] Department of Chemistry, Princeton University, Princeton, NJ, USA. [2] Department of Physics, Princeton University, Princeton, NJ, USA. [3] Program in Applied and Computational Mathematics, Princeton University, Princeton, NJ, USA. [4] Princeton Institute for the Science and Technology of Materials, Princeton University, Princeton, NJ, USA. [5] Department of Chemical and Biological Engineering, Princeton University, Princeton, NJ, USA. ✉email: pdebene@princeton.edu

Among water's many distinctive properties is its complex phase behavior at low temperatures and/or high pressures. Water can solidify into at least 17 different ordered crystalline structures[1], and it also exhibits polyamorphism, i.e., multiple forms of disordered glassy states that transform into each other in apparently discontinuous fashion[2–4]. There are many routes to prepare the various types of amorphous ice[5,6], which differ in their density and local structure. For example, if quenched sufficiently fast to low enough temperatures at atmospheric pressure[7], liquid water forms a glass commonly known as low-density amorphous ice (LDA). Depending on the preparation route, LDA samples can exhibit minor differences in density and/or local structure (e.g., LDA-I and LDA-II)[8]; however, the physical properties of LDA after annealing are largely reproducible[6]. LDA in turn can undergo a pressure- or temperature-driven first-order-like phase transition into high-density amorphous ice (HDA)[4,9]. There are also other forms of HDA ice (e.g., very HDA (VHDA), unannealed HDA, expanded HDA, relaxed HDA)[6], but there is some discussion about whether these high-density polyamorphs are thermodynamically distinct phases or related states that exist along a continuum of structural relaxation[5,6]. The rich and nontrivial nature of water's amorphous solid phases remains an active area of study in both simulation and experiment[5,6,10–12].

In parallel, supercooled liquid water (i.e., water cooled below its melting temperature but maintained in a metastable liquid state) exhibits its own complexity. Computational studies of water-like models have demonstrated that water may undergo a first-order liquid–liquid transition (LLT) into high-density and low-density liquids (HDL and LDL, respectively);[13] the resulting line of liquid–liquid coexistence terminates in a metastable liquid–liquid critical point (LLCP)[14]. The LLT hypothesis posits that such a transition exists in real supercooled water[14]. In this viewpoint, the thermodynamic consequences of the existence of the LLCP can explain many of liquid water's anomalous thermophysical properties. For instance, sharp increases in water's thermodynamic response functions (e.g., isothermal compressibility or heat capacity) are related to enhanced fluctuations, which diverge at the critical point, and show sharp maxima along the so-called Widom line, which is a locus of maximum correlation length extending from the LLCP to higher temperatures and lower pressures[15,16]. While the loci of maximum compressibility, heat capacity, and correlation length are distinct, they become indistinguishable asymptotically close to the critical point and are often referred to under the general designation of the Widom line. LLTs have been experimentally observed in some pure substances (e.g., phosphorous[17], sulfur[18], silicon[19], triphenyl phosphite[20]) and a growing body of evidence supports the existence of an LLCP in supercooled liquid water at positive pressures[21–23]. It should be noted that experiments in this regime are extremely challenging due to rapid ice crystallization. As a result, much work aimed at probing the phase behavior of supercooled water has been done via simulation[13]. Simulations of several classes of water models of varying accuracy and complexity have yielded evidence consistent with an LLT[24–27].

Some researchers have suggested the first-order-like transition between LDA and HDA provides evidence of the LLT in water, with LDA considered to be the structurally arrested analog of LDL and HDA the corresponding analog of HDL[5,28]. The metastable/glassy water phase diagram is sometimes drawn with the LDA/HDA transition line as the direct extension of the LDL/HDL transition line to lower temperatures[29]. While such an analogy is attractive, given the still incompletely understood nature of the glass transition[30], as well as the nontrivial effects of sample preparation procedure on the structure and properties of the ice polyamorphs[12], a direct correspondence between the HDA/LDA and HDL/LDL transitions is still an open question[31].

A number of recent computational studies have noted similarities between the amorphous ices and associated supercooled liquid in both structural characteristics[32,33] and via the potential energy landscape formalism[34,35]. New experiments have also suggested that HDL is directly accessible upon heating HDA and VHDA at elevated pressure[36]. Furthermore, recent compelling experimental evidence in support of the LLT in water relied on the heating-induced pressurization of HDA to form HDL, which then discontinuously transformed to LDL as the pressure relaxed[23]. On the other hand, experiments and simulations have also noted structural commonalities between HDA and crystalline ice IV, suggesting that HDA could be more closely connected to the metastable ice IV polymorph rather than a HDL[37,38]. These recent studies underscore the importance of having a clear picture of the relationship between water's polyamorphism and the possible existence of an underlying LLCP in the supercooled liquid[31,39].

Apart from the aforementioned phase behavior, the amorphous ices also exhibit interesting structural motifs. In particular, computational studies of the TIP4P/2005 water model[40] revealed that LDA and HDA structures were nearly hyperuniform (i.e., exhibiting an anomalous suppression of long-range density fluctuations compared to simple liquids[41]), but the hyperuniformity was broken at the pressure-induced LDA/HDA transition[42]. Apart from the surprising discovery of this uncommon structural signature in disordered states of such a common substance, this work suggested that long-range structure can be used as a powerful metric to track non-equilibrium transformations in glasses. Hyperuniformity can be identified through the long-wavelength limit of the material's static structure factor, $S(k)$[41]. In hyperuniform systems, $S(k) = 0$ as the wavenumber $k$ tends to 0, reflective of the anomalous suppression of long-range density fluctuations. By contrast, as a system approaches a critical point, diverging correlation lengths in the fluid result in long-range density fluctuations that correspond to a diverging $S(k)$ at low $k$ (i.e., an exactly antihyperuniform state). Thus, for a system that exhibits hyperuniform glassy states and an LLCP (e.g., TIP4P/2005[25,42]), hyperuniform and antihyperuniform states could in principle occur in close proximity to each other in the phase diagram. Exploring this intriguing contrast in more detail could be instructive in illuminating the linkages between metastable criticality and polyamorphism, both in water and more generally[31,39].

In this work, we used molecular dynamics simulations to generate glasses in three systems: the TIP4P/2005 classical atomistic water model[40], the coarse-grained mW water model[43], and the binary Kob–Andersen (KA) mixture[44]. TIP4P/2005 successfully captures water's phase behavior and anomalous properties and was recently shown to exhibit an LLCP with critical parameters $T_c = 172$ K and $P_c = 1861$ bar[25]. By contrast, mW exhibits water-like tetrahedral local structures and thermophysical anomalies[43] (including polyamorphism[45]) but crystallizes too quickly to allow the observation of any underlying LLCP (should it exist). Finally, the KA mixture is a canonical glass-former that serves here as a simple liquid reference system. We prepared the amorphous solids by isobaric quenching of the equilibrated liquid at various pressures and examined long-range density fluctuations in the glassy states via the zero-wavenumber limit of the static structure factor, $S(0)$. Interestingly, we observed a strong signature of the metastable LLCP in the long-range structure of TIP4P/2005 glasses; this signature was absent in the mW and KA systems. We also discuss trends in the pressure dependence of the glass transition temperature in these systems, and we compare quantitatively how these systems fall out of equilibrium upon cooling via the non-equilibrium index, which characterizes the relationship between $S(0)$ and the isothermal compressibility[46].

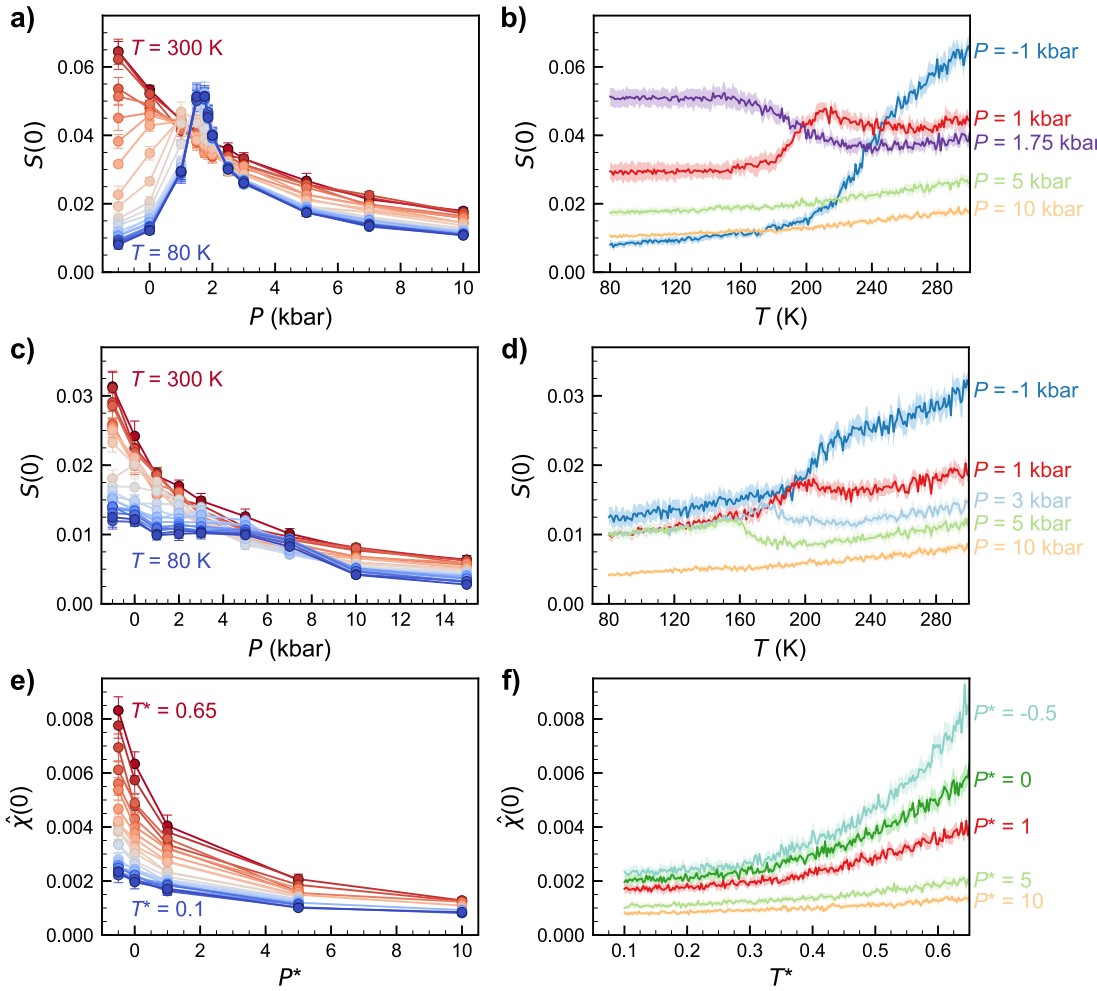

**Fig. 1 Long-range structure during isobaric cooling trajectories at pressure P.** Zero-wavenumber limit of the static structure factor ($S(0)$) for TIP4P/2005 cooled at $q_T = -1.0$ K ns$^{-1}$ (**a**, **b**) and of mW cooled at $q_T = -10.0$ K ns$^{-1}$ (**c**, **d**) and **e**, **f** spectral density ($\hat{\chi}(0)$) for KA at $q_T = -5.55 \times 10^{-6} \tau^{-1}$. In **a**, **c**, **e**, colors denote different temperatures, ranging from $T = 300$ K (red) to $T = 80$ K (blue) in steps of 10 K for **a**, **c** and $T^* = 0.65$ (red) to $T^* = 0.1$ (blue) in steps of 0.0025 for **e**. In **b**, **d**, **f**, colors denote select pressures as marked. Error bars in **a**, **c**, **e** or shaded regions in **b**, **d**, **f** denote 95% confidence intervals obtained from the standard error of the mean.

Our results suggest that long-range structural order metrics bring to light the intricate ties between equilibrium, metastable equilibrium, and non-equilibrium phenomena in water-like models and reveal a subtle-yet-striking relationship between metastable criticality and glass structure. Importantly, our computational results suggest a possible experimental route to identifying the presence of an LLCP in water and other fluids.

## Results

In Fig. 1, we show the long-range structure of the three systems studied in this work, during isobaric cooling trajectories at cooling rates of $q_T = -1.0$ K ns$^{-1}$ (TIP4P/2005), $q_T = -10$ K ns$^{-1}$ (mW), and $q_T = -5.55 \times 10^{-6} \tau^{-1}$ (KA). All quantities related to the KA system are reported in the standard Lennard-Jones (LJ) reduced units scheme of length ($\sigma$), energy ($\varepsilon$), and mass ($m$); if the LJ units for temperature ($T^*$) and time ($\tau$) are converted to real units using the LJ parameters for Argon, the KA $q_T$ corresponds to approximately $-1.0$ K ns$^{-1}$. We chose a $q_T = -10$ K ns$^{-1}$ for mW because this was the smallest cooling rate possible while avoiding crystallization[47]. For the single-component TIP4P/2005 and mW systems (Fig. 1a, b and 1c, d, respectively), we examined long-range density fluctuations via $S(0)$, while for the binary KA system

(Fig. 1e, f) we plot the spectral density ($\hat{\chi}(k)$) at zero wavenumber, an analogous quantity for mixtures that characterizes local volume fraction fluctuations[46,48]. We note that, throughout the text, our use of the term fluctuations refers to spatial fluctuations within a single configuration (i.e., structural inhomogeneities) and not temporal fluctuations; we make this distinction to avoid complications related to the drastically different relaxation time scales in liquids and glasses.

At high temperatures, in the equilibrium liquid state, the three systems showed monotonic trends in long-range density (or volume fraction) fluctuations as a function of pressure (Fig. 1a, c, e), sharply increasing at low and negative pressures as the systems approached the liquid–vapor spinodal and decreasing monotonically as pressure increased. Strikingly, while the KA system (Fig. 1e) retained the same qualitative features throughout the isobaric quench, the water-like TIP4P/2005 model (Fig. 1a) developed a sharp peak at intermediate pressure as the system was cooled past the glass transition. Given that $S(0)$ is reflective of long-range density fluctuations in the material, this peak in $S(0)$ is indicative of increased long-range structural correlations in water-like glasses prepared at intermediate pressure relative to those formed at low or high pressures. The mW system (Fig. 1c) exhibited features somewhat intermediate between KA and

TIP4P/2005. At the lowest temperatures, we observed some small signatures of non-monotonicity in $S(0)$ vs. $P$, but they were much weaker than in TIP4P/2005. This behavior is consistent with the emerging understanding of the mW model, which exhibits many of water's anomalies (e.g., a compressibility maximum upon cooling) but not strongly enough to produce metastable liquid–liquid criticality[49]. Moving our attention to density fluctuations as a function of temperature along a given isobaric cooling ramp, the KA system exhibited a monotonic decrease in $\hat{\chi}(0)$ as the system was cooled for all pressures (Fig. 1f), while both TIP4P/2005 (Fig. 1b) and mW (Fig. 1d) had pressures at which $S(0)$ passed through local minima and/or maxima. Notably, given the relationship between $S(0)$ and the isothermal compressibility[50], in TIP4P/2005 we expect the local maximum in $S(0)$ vs. $T$ at pressures below $P_c$ (e.g., the $P = 1$ kbar curve in Fig. 1b) to be indicative of the system passing the Widom line as it is cooled.

To better understand the anomalous long-range density fluctuations in TIP4P/2005 glasses, we performed analogous isobaric cooling simulations at different cooling rates, ranging from $q_T = -0.1$ K ns$^{-1}$ to $q_T = -10^4$ K ns$^{-1}$, which we plot in Fig. 2. Prior simulation studies with other water models[51–53] have established $q_T$ as an important parameter controlling the structural and energetic properties of water glasses; here we evaluate the effects of $q_T$ on TIP4P/2005's long-range structure. As the cooling rate increased, the peak in $S(0)$ vs. $P$ broadened and shifted to lower pressure. At the highest cooling rate explored, the peak in $S(0)$ disappeared, and the monotonic behavior of the high-temperature liquid was quenched into the glass structure. Interestingly, this is the same cooling rate ($q_T = -10^4$ K ns$^{-1}$) that was previously observed to suppress the density (pressure) anomaly in glasses formed under isochoric conditions, albeit with a different water model[53]. Strikingly, at the slowest cooling rate, the peak in $S(0)$ was nearly coincident with TIP4P/2005's recently identified $P_c = 1861$ bar[25]. Thus, we conjecture that the long-range structure of the TIP4P/2005 glass reflects the fluid's growing correlation length[54,55] at pressures approaching the LLCP.

Within the Ornstein–Zernike formalism for liquids, the S(k) at low k near a critical point can be decomposed into a non-critical background contribution and an anomalous critical contribution that depends on the correlation length of critical fluctuations[25,56]. A two-state interpretation of water's thermodynamics has shown that the background contribution can exhibit a maximum upon cooling, even in systems that lack a critical point[57]. This maximum in the non-critical component of the S(k) could be responsible for the

small feature in the mW $S(0)$ near $P = 5$ kbar (Fig. 1c). By contrast, we interpret the peak in the TIP4P/2005 $S(0)$ vs. $P$ to be a result of critical density fluctuations due to the close correspondence between the location of the peak and TIP4P/2005's unambiguously identified LLCP[25], as well as the relative magnitude of the peak $S(0)$ compared to the high- and low-pressure limits. We attempted an Ornstein–Zernike-like fit[25] to the $S(k)$ to rigorously separate the non-critical and anomalous scattering contributions and characterize the growth of the critical correlation length as a function of $(T, P)$, but we were unable to obtain a unique fit to the $S(k)$ due to numerical noise at low $k$ for the moderate system size considered herein (we plot a representative $S(k)$ in Supplementary Fig. 1). Such an effort (necessitating a larger system or a large number of replicate simulations) would be a worthwhile avenue for future work. Nevertheless, an intriguing implication of the results presented in Fig. 2 is that the non-equilibrium glass structure at $T = 80$ K retains signatures of TIP4P/2005's metastable LLCP, despite the nearly 100 K difference in temperatures between the regimes of interest. In order to rationalize this behavior in the context of the development of the non-equilibrium glassy state upon cooling, we must also characterize the structural arrest of these systems as they undergo the glass transition.

Figure 3 shows the glass transition temperature ($T_g$) for all isobaric cooling trajectories considered in this work, obtained by the intersection of lines fit to the high- and low-temperature branches of the enthalpy vs. $T$ curves along the cooling ramps. For TIP4P/2005 and mW (Fig. 3a, b), we observed a change in slope of $T_g$ vs. $P$ from anomalous negative slope at low pressures to simple liquid-like positive slope at high pressures; this observation qualitatively matches similar computational studies with water-like models[9,58], as well as available experimental data[12]. A positive slope was observed for the KA system (Fig. 3c) at all pressures, as expected. The anomalous minimum in $T_g$ vs. $P$ for water-like models has been shown to be associated with the locus of maximum diffusivity in the fluid as a function of pressure ($D_{max}(P)$)[58]. Furthermore, supercooled water's anomalous dynamics (of which $D_{max}(P)$ is a notable aspect) can be rationalized in terms of water's tendency to form locally favored structures at low temperatures and pressures and a possible crossover from LDL-like to HDL-like liquids[59,60]. In the present work, pressures at which $T_g$ vs. $P$ exhibited a negative slope corresponded to LDA-like[61] structures as characterized by the $S(k)$ and the oxygen–oxygen radial distribution function ($S(k)$ and $g(r)$, see Supplementary Figs. 2 and 3)), while a positive slope corresponded to HDA-like structures[61], and a near-zero slope corresponded to a combination of LDA-like and HDA-like local structures (Supplementary Fig. 3)[62]. This observation lends credence to the structurally based interpretation of water's dynamic anomalies[59,60]. We also calculated other fluid properties at $T_g$, such as density and thermal expansion coefficient, which showed similarly anomalous behavior for pressures below $P_c$; we discuss these trends briefly in Supplementary Fig. 4 and will explore them in more detail in future work.

Furthermore, the trends in $T_g$ in the TIP4P/2005 system showed considerable richness (Fig. 3d). Over the range of $q_T$ explored herein, the locus of $T_g$ vs. $P$ shifted from lying below, to nearly coincident with, to lying above the LLCP (blue X in Fig. 3d), upon increasing the cooling rate. In Fig. 3d, we mark the pressure at which the maximum value of $S(0)$ was observed for each cooling rate (corresponding to the peaks in $S(0)$ vs. $P$ in Fig. 2) with dark symbols connected with dashed lines; this locus of maximum $S(0)$ extends from the LLCP to higher $T$ and lower $P$, much as we would expect for the shape and location of the Widom line and/or locus of maximum compressibility.

Thus, we form the following qualitative picture of the relationship between the non-equilibrium glass structure and the

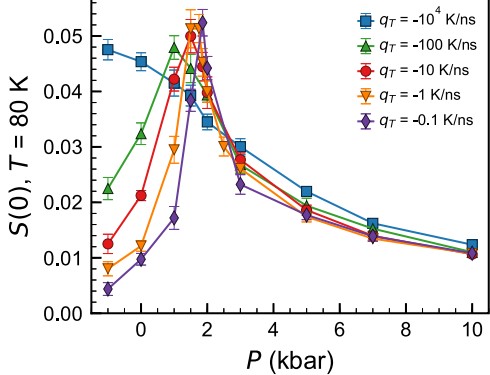

**Fig. 2 Cooling rate effects in TIP4P/2005.** Long-range limit of the static structure factor ($S(0)$) of TIP4P/2005 glasses formed by isobaric cooling to $T = 80$ K at various pressures ($P$). Colors and symbols denote different cooling rates ($q_T$) as marked, and error bars denote 95% confidence intervals obtained from the standard error of the mean.

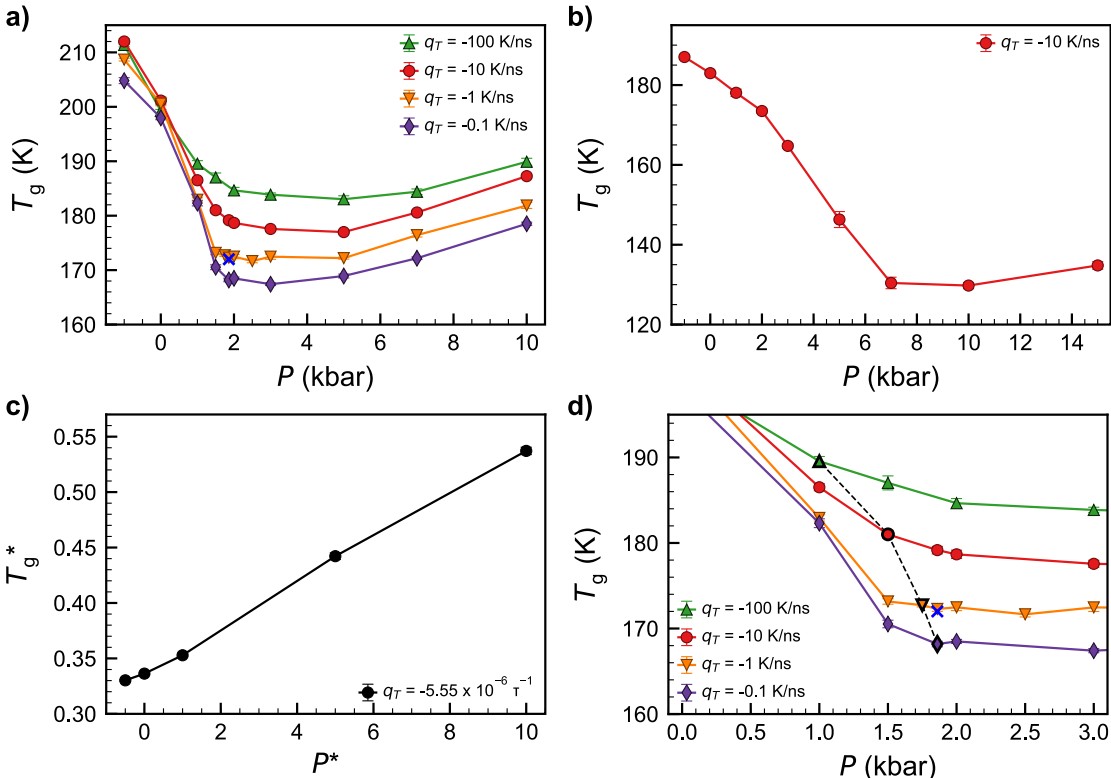

**Fig. 3 Glass transition temperature ($T_g$) as a function of pressure $P$. a**, **d** TIP4P/2005, **b** mW, and **c** Kob–Andersen mixture. In **d**, the $T_g$ vs. $P$ data is the same as in **a** but presented on a different axis scale to focus on the near-critical region; the dashed black line denotes the locus of maximum $S(0)$ from Fig. 2, and the blue X denotes the LLCP location from ref. [25]. In all panels, colors and symbols denote different cooling rates ($q_T$) as marked and error bars denote 95% confidence intervals obtained from the standard error of the mean.

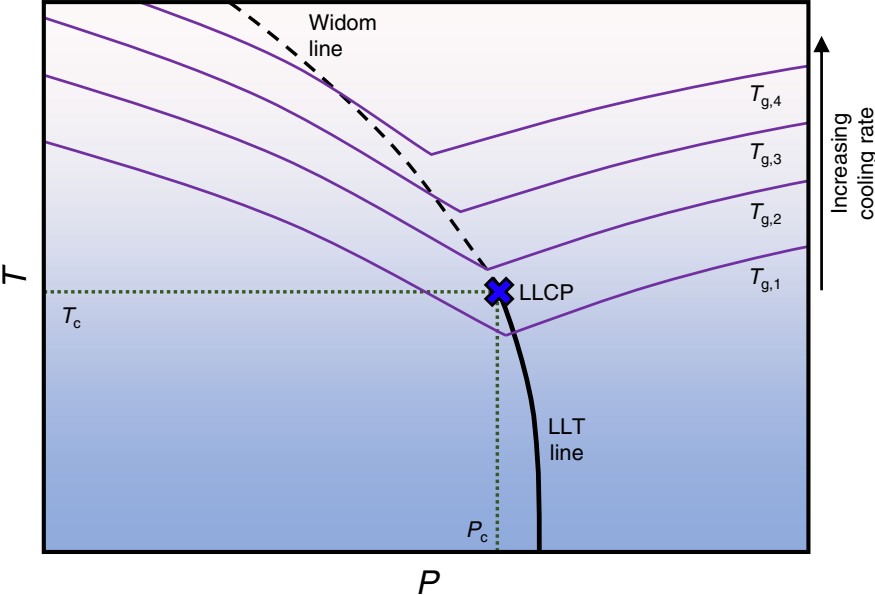

**Fig. 4 Schematic metastable and glassy water phase diagram in the temperature–pressure plane.** Purple lines represent the loci of glass transition temperature ($T_g$) vs. pressure ($P$) for different cooling rates. Black solid and dashed lines represent the LLT and Widom lines, respectively. The LLCP is represented with a blue X.

metastable equilibrium LLCP in TIP4P/2005, which is illustrated schematically in Fig. 4. As the cooling rate increases, the locus of $T_g$ vs. $P$ shifts to higher temperatures. As the liquid undergoes structural arrest in the general region of the glass transition, the $T = 80$ K glass can be thought of as a snapshot of the liquid structure in the vicinity of ($T_g$, $P$). Thus, the trends in $S(0)$ as a function of $P$ and $q_T$ are reflective of the increased correlation length as the system passes the Widom line or LLCP as it vitrifies, and the peak value of $S(0)$ at a given $q_T$ reflects the intersection of the Widom line with the ($q_T$-dependent) locus of $T_g$ vs. $P$. In

other words, the temperature- and pressure-dependent intersection of the Widom line with the arrested dynamics along the locus of $T_g(P, q_T)$ gives rise to the asymmetric shift of the maximum in $S(0)$ to lower $P$ with increasing $q_T$ (Fig. 2). We note, importantly, that the change in slope of the $T_g$ vs. $P$ as the system transitions from LDA-like to HDA-like behavior (Fig. 3a–d and Supplementary Fig. 3) is also broadly coincident with the Widom line (or line of maximum $S(0)$).

As a final illustration of the usefulness of long-range structure in characterizing the systems as they vitrify in the presence/absence of an LLCP, we calculated the non-equilibrium index, $X = \frac{S(0)}{\rho \kappa_T k_B T} - 1$, proposed in ref. [46]. In equilibrium, $S(0)$ is related to the isothermal compressibility of the fluid ($\kappa_T$) via $S(0) = \rho \kappa_T k_B T$[50], in which $\rho$ is the number density of molecules in the fluid and $k_B$ is the Boltzmann constant. Thus, if a system is in thermal equilibrium, $X = 0$, and deviations of $X$ from zero reflect the degree to which the system falls out of equilibrium upon cooling. In previous work, $X$ was found to increase sharply as the system was cooled past the glass transition, whereupon the long-range structure could no longer relax on the time scale imposed by the cooling rate[46]. In Fig. 5, we plot $X$ for the three systems studied in this work, at the same set of cooling rates reported in Fig. 1 (we calculate $\kappa_T$ using the volume fluctuations of the simulation box at a given temperature as described in "Methods"). To facilitate comparison across models and state points, we normalize the temperatures in Fig. 5 by the (pressure-dependent) $T_g(P)$ reported in Fig. 3. For TIP4P/2005, the system showed the expected qualitative behavior at all pressures, in which $X$ was near zero at high temperatures and increased sharply as the system approached $T_g$. However, the slope of said increase was strongly pressure dependent. At those pressures closest to the critical pressure (or the peak value in $S(0)$ for this cooling rate), the magnitude of the $X$ vs. $T$ slope was found to be the largest, i.e., the system fell out of equilibrium more rapidly upon cooling. It thus appears that the $X$ metric captures critical slowing down[63] in which structural relaxation is anomalously slowed near the LLCP. Significantly, the pressures with the highest slope magnitude also corresponded to the pressures at which the $T_g$ vs. $P$ curves (Fig. 3) switched from anomalous to simple liquid behavior and the pressures at which the system transitioned from LDA-dominated to HDA-dominated structures (Supplementary Fig. 3). The mW system exhibited a similar qualitative trend, in which the largest slope in $X$ occurred within a similar range of pressures as the change in slope of $T_g$ vs. $P$ (or the shift from LDA-like to HDA-like structures), but the signal was not as strong as in TIP4P/2005. By contrast, the KA system shows no obvious pressure dependence in $X$. To more directly visualize these trends in the slope of $X$, including the potential impact of critical slowing down in TIP4P/2005 near the critical pressure, we also plot $\left|\frac{dX}{dT}\right|$ in Supplementary Fig. 5. These results suggest that $X$ reflects an interesting connection between dynamic and thermodynamic phenomena in the region of the LLT and glass transitions and reveals another metric by which systems that exhibit water-like anomalies can be distinguished from simple liquids.

To close, we emphasize that the structures of the glasses discussed herein are produced as a complicated combination of the temperature and pressure dependence of the (metastable) equilibrium fluid properties (e.g., density fluctuations, isothermal compressibility) and the temperature, pressure, and cooling rate dependence of the glass transition. For example, if the TIP4P/2005 system were able to reach thermal equilibrium near the LLCP, we would expect a sharp increase in $S(0)$ at temperatures near $T_c$, followed by a decrease in $S(0)$ at lower temperatures.

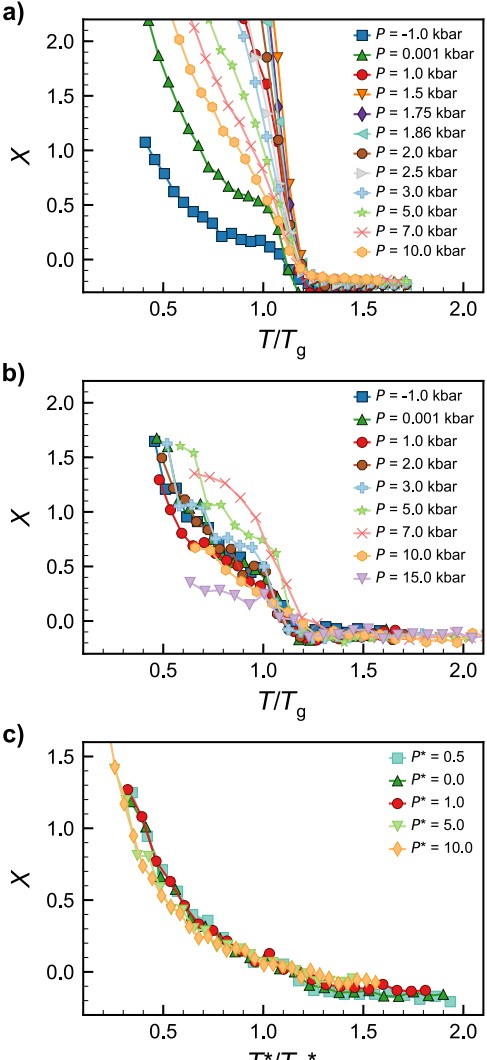

**Fig. 5 Non-equilibrium index ($X$) during isobaric cooling. a** TIP4P/2005 at a cooling rate $q_T = -1.0$ K ns$^{-1}$, **b** mW at $q_T = -10$ K ns$^{-1}$, and **c** Kob–Andersen mixture at $q_T = -5.55 \times 10^{-6}$ $\tau^{-1}$. Colors and symbols denote different pressures as marked. The reported values represent the average value of $X$ over a (**a**, **b**) $T = 10$ K or (**c**) $T^* = 0.025$ temperature window to reduce numerical noise. All temperatures are plotted normalized to the glass transition temperature ($T_g$) at that pressure and cooling rate.

However, considering the $P = 1.75$ kbar isobar in Fig. 1b, $S(0)$ increased modestly as the system approached $T_c = 172$ K and did not evolve further for $T < 160$ K due to the dynamical influence of the glass transition ($T_g \sim 170$ K for that pressure and cooling rate). The critical slowing down phenomenon noted above in the context of the non-equilibrium index also likely plays a role in the degree to which $S(0)$ can increase/decrease in the immediate vicinity of the LLCP. It is also possible that finite size effects could influence the numerical values for $S(0)$ obtained in this work (due to the finite system limiting the wavelength of critical density fluctuations that can develop), but we expect the glass transition to be the major contributor to the lack of structural evolution at low $T$. These observations outline the rich possibilities for future investigations to map in detail the relationship between the various dynamic and thermodynamic phenomena at play in controlling the structure of water glasses.

## Discussion

In this work, we used molecular dynamics simulations to examine the structure of water-like and simple liquid glasses formed under isobaric conditions. For the TIP4P/2005 water model, we observed a strong signature of the LLCP in the form of increased long-range density fluctuations in glasses formed at pressures proximate to the critical point. Interestingly, while such signatures were absent in the $S(0)$ of the coarse-grained mW system, we did see water-like anomalies in mW's glass transition temperature and non-equilibrium index, $X$. This result aligns with the understanding of mW as exhibiting many water-like features but lacking an LLCP. We also note that the signatures of criticality we observed in the TIP4P/2005 glasses were enabled by the (perhaps fortuitous) close proximity of the glass transition temperatures for the cooling rates considered and the location of the LLCP. In light of work that has demonstrated the ability to tune the temperature and relative stability of the LLCP as a function of the angular flexibility of the tetrahedral network structures[64,65], it would be instructive to perform similar tests for systems in which the LLCP occurs somewhat above or somewhat below the glass transition.

Crucially, we note that the approach we follow in this current computational study (i.e., structural characterization via the low-wavenumber limit of the structure factor of glasses formed by isobaric quenching at various pressures) is potentially realizable experimentally via scattering techniques. Thus, this approach could represent another experimental route to add to the growing body of evidence for an LLCP in water[23] and may provide a relatively facile metric by which LLCPs can be identified in other substances.

Of more general relevance to the physics of glasses, we observed that, while in simple liquid systems high pressures are needed to suppress long-range fluctuations in the glass, our results for TIP4P/2005 show that for realistic water-like systems suppressed long-range density fluctuations occur in glasses prepared at both low and high pressures. As such, it would be very interesting to probe in detail the structural evolution upon cooling to examine the differences in glass formation at negative, low, and high pressures for both tetrahedral and simple liquids. Overall, our results demonstrate the utility of tracking long-range structure during glass formation and reveal an interesting relationship between the LLT, polyamorphism, and the glass transition in realistic water-like models. We hope that these results motivate others to further explore the rich non-equilibrium and metastable phase behavior of water and other tetrahedral liquids via investigations of the glass and stimulate both simulation and experimental investigations of glass formation under applied and negative pressure.

## Methods

**Simulation details**. We performed molecular dynamics simulations of glass formation using the TIP4P/2005[40] and mW[43] water models, as well as the KA[44] LJ mixture. We used GROMACS v2018.4[66] for the TIP4P/2005 simulations and LAMMPS v7Aug19[67] for the mW and KA simulations. For TIP4P/2005, we used a leap-frog integrator with a time step size of 2 fs, a stochastic velocity-rescaling thermostat with relaxation time 0.1 ps, and a Parrinello–Rahman barostat with relaxation time 1.0 ps. Bond and angle constraints were enforced with a sixth-order LINCS algorithm, and the Van der Waals and real-space Coulombic interaction cutoffs were 1.2 nm. We used a particle-mesh Ewald scheme to treat long-range electrostatics with a Fourier grid spacing of 0.16 nm. For the mW simulations, we used a velocity-Verlet algorithm with time step size 0.01 ps and the LAMMPS Nose–Hoover-type thermostat and barostat with relaxation times 1 and 10 ps, respectively. For the KA system, we used a time step size of $0.003 \tau$ and thermostat and barostat relaxation times 0.3 and $3.0 \tau$, respectively. The TIP4P/2005 and mW systems had 8192 molecules, and KA system had 8192 type A particles and 2048 type B particles to maintain the standard 80:20 A:B composition[44]. The KA LJ parameters were $\varepsilon_{AA} = 1.0$, $\varepsilon_{BB} = 0.5$, $\varepsilon_{AB} = 1.5$, $\sigma_{AA} = 1.0$, $\sigma_{BB} = 0.88$, $\sigma_{AB} = 0.8$, with the LJ interactions truncated and shifted at $2.5*\sigma_{ij}$ for each pair of particle types $i$ and $j$. For all simulations, we first equilibrated the systems in the liquid phase ($T = 300$ K for all TIP4P/2005 and mW simulations, $T^* = 0.65$ for the KA

mixture at $P^* \leq 5$, and $T^* = 0.85$ for KA at $P^* = 10$) at a given pressure for 30 ns for TIP4P/2005 and mW and $1.2 \times 10^4 \tau$ for KA. Then we cooled the systems in a stepwise fashion with steps of 1 K for TIP4P/2005 and mW and 0.0025 for KA, holding at constant temperature for a given length of time before decreasing instantaneously to the next temperature step to achieve a desired cooling rate, $q_T$. The final temperature for TIP4P/2005 and mW was $T = 80$ K, and the final temperature for KA was $T^* = 0.1$. We used a cubic simulation box with periodic boundary conditions for all systems and ran ten independent replicates at each condition.

**Analysis details**. We computed the static structure factor $S(k)$ or the spectral density $\hat{\chi}(k)$ for the final configuration at each temperature along the cooling ramps. For the TIP4P/2005 and mW systems, we used the following expression

$$S(\mathbf{k}) = \frac{1}{N} \left| \sum_{m=1}^{N} e^{-i\mathbf{k} \cdot \mathbf{r}_m} \right|^2 \tag{1}$$

in which $N$ is the total number of water molecules and $\mathbf{r}_m$ is the vector position of molecule $m$ in the simulation box. The possible set of wavevectors $\mathbf{k}$ is defined by $\mathbf{k} = \frac{2\pi}{L} \left( n_x \hat{\mathbf{x}} + n_y \hat{\mathbf{y}} + n_z \hat{\mathbf{z}} \right)$, where $L$ is the side length of the simulation box; $\hat{\mathbf{x}}$, $\hat{\mathbf{y}}$, and $\hat{\mathbf{z}}$ are the unit vectors in their respective directions; and $n_x$, $n_y$, and $n_z$ run over all integer values. We obtained the one-dimensional $S(k)$ by radially averaging $S(\mathbf{k})$ over each point at a given wavenumber $k = |\mathbf{k}|$. For the TIP4P/2005 system, we take the position of the oxygen atom as a proxy for the position of the molecule (this definition means that the total $S(k)$ reported in this work is also equivalent to the oxygen–oxygen partial structure factor). We extrapolated the calculated $S(k)$ to $k = 0$ by fitting a quadratic function $y = c_2 k^2 + c_0$ to the obtained $S(k)$ over the range $0.2 \, \text{Å}^{-1} \leq k \leq 1.0 \, \text{Å}^{-1}$. We note that an additional anomalous scattering contribution[25] resulted in a low-$k$ uptick in $S(k)$ in our simulations of TIP4P/2005 glasses formed near the LLCP; our quadratic extrapolation procedure does not capture this near-critical contribution and may slightly underestimate the TIP4P/2005 $S(0)$ for pressures and temperatures near criticality. However, we tested several different types of $S(k)$ extrapolation procedures and found that our qualitative conclusions held in all cases, as discussed in detail in Supplementary Figs. 6 and 7. The mW system did not display any signs of a low-$k$ uptick in $S(k)$ for any state points studied; this result supports our assertion that mW does not exhibit an LLCP. We also note that the $S(0)$ values we report in this work for TIP4P/2005 are approximately a factor of two higher than the $S(0)$ reported in ref. [42] for LDA configurations prepared under the same conditions. We posit that this discrepancy is due to differences in the set of $k$-vectors and $S(k)$ extrapolation procedures used; however, this numerical detail does not impact the key qualitative conclusions described herein or in ref. [42].

$\hat{\chi}(k)$ is an analogous quantity to $S(k)$ and characterizes local volume fraction fluctuations in mixtures[46,48]. We calculated $\hat{\chi}(k)$ for the KA system via the expression given in Eq. 34 of ref. [46] and extrapolated to $k = 0$ by fitting a fourth-order polynomial $y = c_4 k^4 + c_2 k^2 + c_0$ over the range $0.6 \, \sigma^{-1} \leq k \leq 3.0 \, \sigma^{-1}$. See Supplementary Figs. 6 and 8 for examples of our fitting and extrapolation procedure for both $S(k)$ and $\hat{\chi}(k)$.

We calculated the glass transition temperature, $T_g$, by fitting lines to the high- and low-temperature branches of the enthalpy vs. temperature curves along the cooling ramps (see Supplementary Fig. 9 for an example of our $T_g$ construction). We defined the enthalpy at a given $T$ as the average value of the enthalpy over the second half of each temperature step. We calculated the non-equilibrium index, $X$[46], in the single-component systems as

$$X = \frac{S(0)}{\rho \kappa_T k_B T} - 1 \tag{2}$$

in which $\rho$ is the number density of molecules and $\kappa_T$ is the isothermal compressibility obtained via $\kappa_T = \left( \langle V^2 \rangle - \langle V \rangle^2 \right) / k_B T \langle V \rangle$ over the second half of each temperature step. In the KA system, we adjusted the expression to account for the binary mixture[46,50]

$$X = \frac{S_{AA}(0) S_{BB}(0) - S_{AB}(0)^2}{\kappa_T k_B T \left[ \rho_B S_{AA}(0) + \rho_A S_{BB}(0) - 2 (\rho_A \rho_B)^{0.5} S_{AB}(0) \right]} - 1 \tag{3}$$

in which $S_{AA}$ and $S_{BB}$ are the partial structure factors of species A and B obtained by limiting Eq. 1 to only consider the given species of interest. We obtained $S_{AB}$ by

$$S_{AB}(\mathbf{k}) = \frac{1}{\sqrt{N_A N_B}} \left( \sum_{m=1}^{N_A} e^{-i\mathbf{k} \cdot \mathbf{r}_m} \right) \left( \sum_{n=1}^{N_B} e^{-i\mathbf{k} \cdot \mathbf{r}_n} \right)^* \tag{4}$$

in which the $*$ denotes the complex conjugate of the quantity in the parenthesis and $N_i$ is the number of particles of species $i$. For the partial structure factors in Eqs. 3 and 4, we performed the angular averaging and extrapolation to zero wavenumber in the same manner as described above for the total structure factors. For the plots in Fig. 5, we averaged $X$ over ten consecutive temperature steps in order to reduce numerical noise. In all other plots, error bars are 95% confidence intervals obtained by averaging over the 10 independent replicates.

## Data availability

All data related to this work, including raw simulation trajectory data and processed data used to create all figures in the manuscript, are available for download at the Princeton DataSpace repository at https://doi.org/10.34770/8v5g-b259[68].

## Code availability

All code used in this work, including simulation input files and analysis scripts used to process raw data and create all figures in the manuscript, are available for download at the Princeton DataSpace repository at https://doi.org/10.34770/8v5g-b259[68].

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

## Acknowledgements
The work of T.E.G., S.T., and R.C. was supported by the "Chemistry in Solution and at Interfaces" (CSI) Center funded by the U.S. Department of Energy Award DE-SC001934. P.G.D. acknowledges support from the National Science Foundation (grant CHE-1856704). Computational resources were provided by Terascale Infrastructure for Groundbreaking Research in Engineering and Science (TIGRESS) at Princeton University and the National Energy Research Scientific Computing Center (NERSC), a U.S. Department of Energy Office of Science User Facility operated under Contract No. DE-AC02-05CH11231.

## Author contributions
P.G.D. and T.E.G. conceived of the project; T.E.G. performed research; T.E.G., P.G.D., S.T., and R.C. designed research, discussed results, and wrote the paper.

## Competing interests
The authors declare no competing interests.
