## [Peer Review File · Nature Communications]

REVIEWER COMMENTS

Reviewer #1 (Remarks to the Author):

In the manuscript "Manifestations of metastable criticality in the long-range structure of model water glasses", Gartner et al simulate, using classical molecular dynamics, the cooling of liquid water (TIP4P/2005 and mW) and of KA at different pressures and cooling rates. At each thermodynamic point, the authors compute the value of $S(0)$ and the 'non-equilibrium index' X . Interestingly, the authors find a pronounced increase in the long-range density fluctuations in the vicinity of the pressure associated with the liquid-liquid transition in TIP4P/2005, while no such increase is present in other glasses that, in the liquid phase, do not show a LLT. The authors also find that water-like systems exhibit a strong pressure dependence on the non-equilibrium index, that instead does not show such dependence in simple liquids. Based on these results, the authors claim the existence of a link between metastable equilibrium phenomena of the LLC and non-equilibrium long-range structures in glassy water. Moreover, the authors claim that the protocol described in this manuscript can potentially be implemented in experiments as another route to study supercooled liquids and locate a LLC.

Overall, the paper is well written, and the results are interesting. The introduction of the reported protocol to study supercooled and glassy water is novel and the signatures of the LLT in the $S(0)$ in the glassy states adds new understanding on the field of water. In my opinion, the manuscript deserves to be published in Nature Communications after the author address the following points.

Comments:

1. The authors should mention the following papers when mentioning that the T_g -dependence over the cooling rate has been already investigated with another model of water: [Giovambattista et al "glass-transition temperature of water: a simulation study, PRL 93, 047801 (2004)" and Giovambattista et al "Cooling rate, heating rate, and aging effects in glassy water, PRE 99, 050201(R) (2004)"].
2. Figure 3d & 4: one would expect the LLC to be close to the slowest cooling rate (0.1K/ns). Why, instead, it is closer to the second slower cooling rate (1K/ns)?
3. The non-equilibrium index is useful but not very handy. One is restricted to very large simulation cells to properly sample $S(0)$. Moreover, this index does not capture the dependence on the pressure (Figure 5). One needs to introduce info from the slope of $T_g(P)$. The authors should comment more on this.
4. In the Introduction, the authors list different plausible phases of HDA, but only one phase of LDA. The authors should mention, e.g., LDA_I and LDA_II. Moreover, considering that the authors discuss about "structural motifs" in amorphous ices, they should mention the connections between HDA and the metastable ice IV (see, eg. Shephard et al "Is high-density amorphous ice simply a derailed state along the ice I to ice IV pathway?" J Phys Chem Lett, 8 1645-1650 (2017), and Martelli et al "Searching for crystal like domains in amorphous ices" PRM, 2 075601 (2018)).

Reviewer #2 (Remarks to the Author):

The present work makes evident the existence of long-range density fluctuations for a realistic model (TIP4P/2005) of water. This fact represents a clear signature of the existence of the proposed liquid-liquid critical point (LLCP) for this water system (the existence of a LLC in water represents an issue of major concern within this field and has promoted intense research efforts during the last decades). This behavior, in turn, is found by the authors to be absent in other two model systems that lack a LLC. The work also finds a link between LLC metastability and the non-equilibrium long-range structure of glassy water, while additionally suggesting an experimental way to probe the existence of the LLC.

I find the work to be solid, of high quality, to address a subject of pivotal current concern and to provide results of great relevance of interest for a wide community of researchers across the physical sciences. Thus, I am glad to recommend publication. Nonetheless, there is only one point I would like to raise for the authors to comment: In Fig. 1 a, a very notable extremely sharp peak develops in $S(0)$ around the critical pressure as temperature is lowered. This behavior stems from the increase in the long-range density fluctuations and

it would be expected to grow as we approach the critical temperature, T_C , where fluctuations should be maximal (diverge if at the critical point). However, when going down in temperature from around T_C to the lower temperatures studied, the curves clearly superimpose each other. Is this a sign of the presence of finite size effects? Does the system size become small in comparison with the range of the fluctuations?

Reviewer #3 (Remarks to the Author):

In this paper, the authors report an interesting numerical simulation observation that the amorphous state formed by temperature cooling under various pressures remembers the critical fluctuations that water experiences during the cooling process. The authors used a model water, TIP4P/2005, clearly proven to have a second critical point. As a result, long-range density correlations were observed in the amorphous state obtained by temperature cooling at pressures near the critical pressure. These results of the TIP4P/2005 model were compared with those of the mW water and Kob-Andersen binary mixtures. The results show that the peculiar behavior observed in the TIP4P/2005 model is not observed at all in the KA model, which does not have a second critical point. On the other hand, the mW model results show a weak signal indicating the enhancement of density fluctuations at a pressure near the $T_g(P)$'s minimum.

The enhancement of long-range density correlations in the glassy state results from an intriguing combination of criticality and slow glassy dynamics. This stems from the special relationship between the critical-point location and the glass-transition line in the water's T-P phase diagram. Although it may be technically challenging, it would be possible to apply this strategy to experiments, as suggested by the authors. This unique strategy may provide a new way to detect the second critical point of water, which has been difficult to prove experimentally. This is the first systematic study of the effect of critical fluctuations on water's glassy state to the best of my knowledge.

Given that the criticality of water associated with the liquid-liquid transition has received considerable attention in the community, this manuscript will significantly impact the field and also stimulate experimental research. Thus, I warmly support its publication in Nature Communications. However, before it is accepted, I recommend that the authors consider the following points.

(1) First of all, $S(0)$ is proportional to the isothermal compressibility. Then, the isothermal compressibility is composed of the non-critical background part and the critical part. The high-temperature value of $S(0)$ reflects the background part. For water, it is known that even the non-critical part originating from the two-state feature can have a maximum with decreasing temperature (see, e.g., Fig. 12 of Ref. A). Thus, there is no one-to-one correspondence between the correlation length of critical fluctuations and $S(0)$, as long as the critical contribution is not dominant. In principle, these two contributions can be separated by the detailed analysis of $S(k)$, but which seems difficult in the present case (see Fig. S4). I recommend the authors mention that there can be these two contributions to $S(0)$.

(2) Since it has been proven to have a second critical point for TIP4P/2005 water, the observed $S(0)$ peak as a function of P probably comes from critical fluctuations, as the authors claimed. In contrast, the small peak observed for mW water may not come from the criticality but from the non-critical part's increase due to the two-state feature. A simple two-state model without criticality predicts that the non-critical compressibility peak height as a function of T should monotonically increase with an increase in P (see, e.g., Fig. 12 of Ref. A). On the other hand, T_g decreases with P , which may induce $S(0)$'s quicker decay for a higher P side. The competition between these two tendencies may explain a small $S(0)$ peak around 6 kbar observed for the mW model. However, to draw a definite conclusion, the relationship between the cooling rate and the structural relaxation rate is necessary. There is also a possibility that the second critical point is hidden in the glass state. I recommend the authors discuss these issues briefly.

(3) For density fluctuations to be frozen in glass, they need to grow as they approach the critical point and freeze by glassiness upon further cooling. Therefore, the P -dependence of $S(0)$ may reflect the relationship of the T , P -dependence of the correlation length and lifetime of the density fluctuations with the T , P -dependence of the glass transition. The systematic shift of the $S(0)$ peak to the lower pressure side with

increasing cooling rate, i.e., the asymmetric pressure dependence of the $S(0)$ peak, may also reflect the T_g 's pressure dependence. The dynamic aspect of the origin of the asymmetry deserves some more discussion.

(4) As shown in Fig. 4, T_g decreases as the cooling rate decreases. At the lowest cooling rate $q_T = -0.1$ K/ns, $T_g(P)$ lies below T_c . Thus, one would expect a sharp increase of $S(0)$ peak at the critical pressure at this cooling rate. In Fig. 2, however, we can see only a slight increase of the $S(0)$ peak as the cooling rate decreases. This seems to indicate that critical fluctuations cannot fully develop due to the rapid increase in the lifetime of critical fluctuations in the critical point vicinity. It suggests an interesting competition among the critical slowing down towards T_c , the slowing down of the glassy dynamics towards T_g , and the cooling rate. I recommend the authors discuss this point in more detail.

(5) It would be useful for readers if the authors could provide a rough estimate of the correlation length of density fluctuations. According to Fig. S4, it looks like about $5A$ if we use the Ornstein-Zernike-like analysis (although the lengthscale might be too short for the OZ analysis to be valid).

(6) On page 11, the authors mention the relationship between the minimum of T_g and the maximum diffusivity around 2 kbar. I want to point out that there is another possibility in this regard. Recent studies have shown that the diffusivity maximization around 2 kbar, which is observed even at high temperatures well above T_g , is not a consequence of glassiness but due to the dependence of the activation energy of the liquid on the degree of tetrahedral ordering (see Ref. B). A similar conclusion was also derived recently (see Ref. C). I agree that there is a relationship between the maximum diffusivity and the minimum T_g , but the former may not be a direct consequence of the glassiness. It seems natural to assume that both are due to structural crossover from LDL-like to HDL-like liquids. This scenario may also explain the similar behavior observed for the mW water.

(7) The analysis of the nonequilibrium index X is quite interesting. The increase of dX/dT near the critical pressure clearly indicates the importance of the critical slowing down. I recommend the authors add plots of dX/dT near the onset of the X increase to Fig. 5.

(8) Recently, it has been shown that the first sharp diffraction peak in the structure factor of water may serve as a good descriptor for characterizing the LDL-like and HDL-like structures (see Ref. D; see also Ref. E on the systematic change of $S(k)$). So, plotting $S(k)$ together with $g(r)$ in Figs. S1 and S2 would be useful for readers.

References

References

[A] H. Tanaka, *Eur. Phys. J. E* 35, 113 (2011).

[B] R. Shi, J. Russo, H. Tanaka, *PNAS*, 115, 9444 (2018).

[C] R. Horstmann and M. Vogel, *J. Chem. Phys.*, 154, 054502 (2021).

[D] R. Shi and H. Tanaka, *JACS*, 142, 2868 (2020).

[E] F. Perakis, et al., *PNAS*, 114, 8193 (2017).

RESPONSE TO REVIEWS

Manuscript ID: NCOMMS-21-01760-T

Title: “Manifestations of metastable criticality in the long-range structure of model water glasses”

Authors: Thomas E. Gartner III, Salvatore Torquato, Roberto Car, and Pablo G. Debenedetti

We thank you for your time in considering our manuscript, your thorough comments and helpful suggestions, and your overall positive evaluation of the manuscript. We have edited the manuscript to address your comments/suggestions, which are summarized in our responses below. We have also prepared a version of the manuscript with our changes highlighted in yellow, which is included with our submission of the revision.

Reviewer 1:

In the manuscript “Manifestations of metastable criticality in the long-range structure of model water glasses”, Gartner et al simulate, using classical molecular dynamics, the cooling of liquid water (TIP4P/2005 and mW) and of KA at different pressures and cooling rates. At each thermodynamic point, the authors compute the value of $S(0)$ and the ‘non-equilibrium index’ X . Interestingly, the authors find a pronounced increase in the long-range density fluctuations in the vicinity of the pressure associated with the liquid-liquid transition in TIP4P/2005, while no such increase is present in other glasses that, in the liquid phase, do not show a LLT. The authors also find that water-like systems exhibit a strong pressure dependence on the non-equilibrium index, that instead does not show such dependence in simple liquids. Based on these results, the authors claim the existence of a link between metastable equilibrium phenomena of the LLC and non-equilibrium long-range structures in glassy water. Moreover, the authors claim that the protocol described in this manuscript can potentially be implemented in experiments as another route to study supercooled liquids and locate a LLC.

Overall, the paper is well written, and the results are interesting. The introduction of the reported protocol to study supercooled and glassy water is novel and the signatures of the LLT in the $S(0)$ in the glassy states adds new understanding on the field of water. In my opinion, the manuscript deserves to be published in Nature Communications after the author address the following points.

RESPONSE: We thank the reviewer for the summary of our findings and the recommendation to publish.

Comments:

1. The authors should mention the following papers when mentioning that the T_g -dependence over the cooling rate has been already investigated with another model of water: [Giovambattista et al “glass-transition temperature of water: a simulation study, PRL 93,

047801 (2004)” and Giovambattista et al “Cooling rate, heating rate, and aging effects in glassy water, PRE 99, 050201(R) (2004)”].

RESPONSE: We have added these relevant references to our discussion of cooling rate effects on pages 9-10:

“Prior simulation studies with other water models⁵¹⁻⁵³ have established q_T as an important parameter controlling the structural and energetic properties of water glasses; here, we evaluate the effects of q_T on TIP4P/2005’s long-range structure.”

2. Figure 3d & 4: one would expect the LLCPC be close to the slowest cooling rate (0.1K/ns). Why, instead, it is closer to the second slower cooling rate (1K/ns)?

RESPONSE: We presume that the reviewer is referring to the idea that the slowest cooling rate should exhibit closest to ‘equilibrium-like’ behavior, hence the suggestion that the T_g obtained using the slowest cooling rate should be closest to the LLCPC. However, there is no requirement that the glass transition in the limit of slow cooling be near to the LLCPC, given that the two phenomena come from distinct microscopic origins. Indeed, [Giovambattista et al., *Sci. Rep.* **2**, 390 (2012)] shows that, for the ST2 model, the T_g (achieved at a finite cooling rate) lies somewhat below the LLCPC, and thus the locus of $T_g(P)$ intersects with the liquid-liquid transition line for that model. By contrast, in the present work with TIP4P/2005, we were able to achieve $T_g(P)$ that lie both below and above the LLCPC by changing the cooling rate. This observation was the basis of our suggestion for future work to test a similar glass formation protocol for systems where the LLCPC is significantly above or below the locus of $T_g(P)$.

3. The non-equilibrium index is useful but not very handy. One is restricted to very large simulation cells to properly sample $S(0)$. Moreover, this index does not capture the dependence on the pressure (Figure 5). One needs to introduce info from the slope of $T_g(P)$. The authors should comment more on this.

RESPONSE: We agree with the reviewer that any metric involving $S(0)$ (including but not limited to the non-equilibrium index) is difficult to sample, especially for the wide parameter space explored in this work. Though challenging to calculate, the non-equilibrium index (X) was especially useful in this work, as it revealed subtle but important features such as the impact of critical slowing down in TIP4P/2005, as noted below by Reviewer #3. We also respectfully disagree with the reviewer’s comment that X does not capture pressure dependence. Indeed, the key observation of Figure 5 is that the slope of X is strongly dependent on the pressure in water-like models, whereas it is insensitive to pressure in the Kob-Andersen mixture. We also note that we account for the effect of pressure by normalizing the x-axis of Figure 5 by the (pressure-dependent) $T_g(P)$ reported in Figure 3. To clarify this point, we have added the following comment to page 13-14:

“To facilitate comparison across models and state points, we normalize the temperatures in Figure 5 by the (pressure-dependent) $T_g(P)$ reported in Figure 3.”

4. In the Introduction, the authors list different plausible phases of HDA, but only one phase of LDA. The authors should mention, e.g., LDA_I and LDA_II. Moreover, considering that the authors discuss about “structural motifs” in amorphous ices, they should mention the connections between HDA and the metastable ice IV (see, eg. Shephard et al “Is high-density amorphous ice simply a derailed state along the ice I to ice IV pathway?” J Phys Chem Lett, 8 1645-1650 (2017), and Martelli et al “Searching for crystal like domains in amorphous ices” PRM, 2 075601 (2018)).

RESPONSE: We thank the reviewer for suggesting these additional clarifications about the complex phase behavior and structure of the amorphous ices. We have added the following sentences to the introduction and included the suggested references:

Page 3: “Depending on the preparation route, LDA samples can exhibit minor differences in density and/or local structure (e.g., LDA-I and LDA-II),⁸ however the physical properties of LDA after annealing are largely reproducible.^{6”}

Page 5: “On the other hand, experiments and simulations have also noted structural commonalities between HDA and crystalline ice IV, suggesting that HDA could be more closely connected to the metastable ice IV polymorph rather than a high-density liquid.^{37,38”}

Reviewer 2:

The present work makes evident the existence of long-range density fluctuations for a realistic model (TIP4P/2005) of water. This fact represents a clear signature of the existence of the proposed liquid-liquid critical point (LLCP) for this water system (the existence of a LLCP in water represents an issue of major concern within this field and has promoted intense research efforts during the last decades). This behavior, in turn, is found by the authors to be absent in other two model systems that lack a LLCP. The work also finds a link between LLCP metastability and the non-equilibrium long-range structure of glassy water, while additionally suggesting an experimental way to probe the existence of the LLCP.

I find the work to be solid, of high quality, to address a subject of pivotal current concern and to provide results of great relevance of interest for a wide community of researchers across the physical sciences. Thus, I am glad to recommend publication.

RESPONSE: We thank the reviewer for the positive evaluation of our work.

Nonetheless, there is only one point I would like to raise for the authors to comment: In Fig. 1 a, a very notable extremely sharp peak develops in $S(0)$ around the critical pressure as temperature is lowered. This behavior stems from the increase in the long-range density fluctuations and it would be expected to grow as we approach the critical temperature, T_C , where fluctuations should be maximal (diverge if at the critical point). However, when going down in temperature from around T_C to the lower temperatures studied, the curves clearly superimpose each other. Is this a sign of the presence of finite size effects? Does the system size become small in comparison with the range of the fluctuations?

RESPONSE: We do believe that it is important to consider finite size effects when interpreting the behavior we observed in this work--the reviewer correctly notes that in the immediate vicinity of the critical point density fluctuations will grow to length scales comparable to the system size. However, in the particular case studied here, we feel that the behavior that the reviewer is noting (the $S(0)$ not truly diverging and eventually ceasing to evolve below a certain temperature) to be more of a dynamic consequence of the non-equilibrium cooling of these systems, rather than limitations of system size. For temperatures near to or below the glass transition, large-scale structural rearrangements become increasingly infrequent, thus largely 'freezing-in' the $S(0)$ for temperatures below a given point. If, instead of the isobaric cooling performed in this work, we were able to fully equilibrate the system at state points near to or below the LLC, we would expect to see a much sharper increase in $S(0)$ followed by a decrease with decreasing temperature, as this reviewer states. This point also relates to the issues raised by Reviewer 3 below in terms of underscoring the important relationship between thermodynamic and dynamic phenomena in controlling the structural evolution in these systems. We have added an additional paragraph of commentary to page 16-17, as well as a few comments throughout the manuscript to draw attention to this point--these additions are reproduced in our responses to Reviewer 3 below.

Reviewer 3:

In this paper, the authors report an interesting numerical simulation observation that the amorphous state formed by temperature cooling under various pressures remembers the critical fluctuations that water experiences during the cooling process. The authors used a model water, TIP4P/2005, clearly proven to have a second critical point. As a result, long-range density correlations were observed in the amorphous state obtained by temperature cooling at pressures near the critical pressure. These results of the TIP4P/2005 model were compared with those of the mW water and Kob-Andersen binary mixtures. The results show that the peculiar behavior observed in the TIP4P/2005 model is not observed at all in the KA model, which does not have a second critical point. On the other hand, the mW model results show a weak signal indicating the enhancement of density fluctuations at a pressure near the $T_g(P)$'s minimum.

The enhancement of long-range density correlations in the glassy state results from an intriguing combination of criticality and slow glassy dynamics. This stems from the special relationship between the critical-point location and the glass-transition line in the water's T-P phase diagram. Although it may be technically challenging, it would be possible to apply this strategy to experiments, as suggested by the authors. This unique strategy may provide a new way to detect the second critical point of water, which has been difficult to prove experimentally. This is the first systematic study of the effect of critical fluctuations on water's glassy state to the best of my knowledge.

Given that the criticality of water associated with the liquid-liquid transition has received considerable attention in the community, this manuscript will significantly impact the field and also stimulate experimental research. Thus, I warmly support its publication in Nature Communications. However, before it is accepted, I recommend that the authors consider the following points.

RESPONSE: We thank the reviewer for their detailed and insightful comments, and their positive review of our work.

(1) First of all, $S(0)$ is proportional to the isothermal compressibility. Then, the isothermal compressibility is composed of the non-critical background part and the critical part. The high-temperature value of $S(0)$ reflects the background part. For water, it is known that even the non-critical part originating from the two-state feature can have a maximum with decreasing temperature (see, e.g., Fig. 12 of Ref. A). Thus, there is no one-to-one correspondence between the correlation length of critical fluctuations and $S(0)$, as long as the critical contribution is not dominant. In principle, these two contributions can be separated by the detailed analysis of $S(k)$, but which seems difficult in the present case (see Fig. S4). I recommend the authors mention that there can be these two contributions to $S(0)$.

RESPONSE: We thank the reviewer for mentioning this important clarification. Indeed, we attempted an Ornstein-Zernike-like fit to the $S(k)$ to rigorously separate the critical and non-critical contributions, but were not able to do so within the numerical accuracy of the $S(k)$ calculated in this work. This would be a worthwhile avenue for future careful study. We have added the following sentences to pages 10-11, and also added a new plot of the $S(k)$ for $P \sim P_c$ to the Supporting Information:

“We attempted an Ornstein-Zernike-like fit to the $S(k)$ to rigorously separate the non-critical and anomalous scattering contributions and characterize the growth of the critical correlation length as a function of (T, P) , but we were unable to obtain a unique fit to the $S(k)$ due to numerical noise at low- k for the moderate system size considered herein (we plot a representative $S(k)$ in SI Figure S1). Such an effort (necessitating a larger system or a large number of replicate simulations) would be a worthwhile avenue for future work.”

(2) Since it has been proven to have a second critical point for TIP4P/2005 water, the observed $S(0)$ peak as a function of P probably comes from critical fluctuations, as the authors claimed. In contrast, the small peak observed for mW water may not come from the criticality but from the non-critical part's increase due to the two-state feature. A simple two-state model without criticality predicts that the non-critical compressibility peak height as a function of T should monotonically increase with an increase in P (see, e.g., Fig. 12 of Ref. A). On the other hand, T_g decreases with P , which may induce $S(0)$'s quicker decay for a higher P side. The competition between these two tendencies may explain a small $S(0)$ peak around 6 kbar observed for the mW model. However, to draw a definite conclusion, the relationship between the cooling rate and the structural relaxation rate is necessary. There is also a possibility that the second critical point is hidden in the glass state. I recommend the authors discuss these issues briefly.

RESPONSE: We agree and thank the reviewer for their interpretation of the possible thermodynamic origin of the weak features in the mW $S(0)$. Reflecting the points raised in this comment, we have added the following sentences to the text on page 10, and added a citation to Ref. A:

“Within the Ornstein-Zernike formalism for liquids, the $S(k)$ at low- k near a critical point can be decomposed into a non-critical background contribution and an anomalous critical contribution that depends on the correlation length of critical fluctuations.^{25,56} A two-state interpretation of water’s thermodynamics has shown that the background contribution can exhibit a maximum upon cooling, even in systems that lack a critical point.⁵⁷ This maximum in the non-critical component of the $S(k)$ could be responsible for the small feature in the mW $S(0)$ near $P = 5$ kbar (Figure 1c). By contrast, we interpret the peak in the TIP4P/2005 $S(0)$ vs. P to be a result of critical density fluctuations due to the close correspondence between the location of the peak and TIP4P/2005’s unambiguously identified LLCPC,²⁵ as well as the relative magnitude of the peak $S(0)$ compared to the high- and low-pressure limits.”

(3) For density fluctuations to be frozen in glass, they need to grow as they approach the critical point and freeze by glassiness upon further cooling. Therefore, the P-dependence of $S(0)$ may reflect the relationship of the T, P-dependence of the correlation length and lifetime of the density fluctuations with the T, P-dependence of the glass transition. The systematic shift of the $S(0)$ peak to the lower pressure side with increasing cooling rate, i.e., the asymmetric pressure dependence of the $S(0)$ peak, may also reflect the T_g 's pressure dependence. The dynamic aspect of the origin of the asymmetry deserves some more discussion.

RESPONSE: We agree with the reviewer’s interpretation on this point. Indeed, in the discussion of Figure 4 (pages 13-14), we argue that the relationship between the (T , P)-dependence of the fluid properties and the (P , cooling rate)-dependence of the glass transition is the source of the complicated trends we observe in this work. We have added an additional sentence to the paragraph on pages 13-14 to make this point more explicit, as well as an additional paragraph of discussion at the end of the results section (see the response to point #4 below):

“In other words, the temperature- and pressure-dependent intersection of the Widom line with the arrested dynamics along the locus of $T_g(P, q_T)$ gives rise to the asymmetric shift of the maximum in $S(0)$ to lower P with increasing q_T (Figure 2).”

(4) As shown in Fig. 4, T_g decreases as the cooling rate decreases. At the lowest cooling rate $q_T = -0.1$ K/ns, $T_g(P)$ lies below T_c . Thus, one would expect a sharp increase of $S(0)$ peak at the critical pressure at this cooling rate. In Fig. 2, however, we can see only a slight increase of the $S(0)$ peak as the cooling rate decreases. This seems to indicate that critical fluctuations cannot fully develop due to the rapid increase in the lifetime of critical fluctuations in the critical point vicinity. It suggests an interesting competition among the critical slowing down towards T_c , the slowing down of the glassy dynamics towards T_g , and the cooling rate. I recommend the authors discuss this point in more detail.

RESPONSE: We again share the reviewer’s interpretation of these results, and the intriguing competition between equilibrium fluid properties and the dynamical aspects of the glass transition are exactly what we hope to highlight in this work. We have added the following paragraph of

discussion to pages 16-17 to expand on these points, which also relates to this reviewer's point #3 above and the comment/question from Reviewer 2:

“To close, we emphasize that the structures of the glasses discussed herein are produced as a complicated combination of the temperature- and pressure-dependence of the (metastable) equilibrium fluid properties (e.g., density fluctuations, isothermal compressibility) and the temperature-, pressure-, and cooling rate-dependence of the glass transition. For example, if the TIP4P/2005 system were able to reach thermal equilibrium near the LLC, we would expect a sharp increase in $S(0)$ at temperatures near T_c , followed by a decrease in $S(0)$ at lower temperatures. However, considering the $P = 1.75$ kbar isobar in Figure 1b, $S(0)$ increases modestly as the system approaches $T_c = 172$ K, and does not evolve further for $T < 160$ K due to the dynamical influence of the glass transition ($T_g \sim 170$ K for that pressure and cooling rate). The critical slowing down phenomenon noted above in the context of the non-equilibrium index also likely plays a role in the degree to which $S(0)$ can increase/decrease in the immediate vicinity of the LLC. It is also possible that finite size effects could influence the numerical values for $S(0)$ obtained in this work (due to the finite system limiting the wavelength of critical density fluctuations that can develop), but we expect the glass transition to be the major contributor to the lack of structural evolution at low- T . These observations outline the rich possibilities for future investigations to map in detail the relationship between the various dynamic and thermodynamic phenomena at play in controlling the structure of water glasses.”

(5) It would be useful for readers if the authors could provide a rough estimate of the correlation length of density fluctuations. According to Fig. S4, it looks like about 5Å if we use the Ornstein-Zernike-like analysis (although the lengthscale might be too short for the OZ analysis to be valid).

RESPONSE: We agree that this suggestion would be a worthwhile analysis, but we were unable to achieve a trustworthy fit using the OZ expression for $S(k)$ due to numerical noise at low- k , as mentioned in the response to point #1 above.

(6) On page 11, the authors mention the relationship between the minimum of T_g and the maximum diffusivity around 2 kbar. I want to point out that there is another possibility in this regard. Recent studies have shown that the diffusivity maximization around 2 kbar, which is observed even at high temperatures well above T_g , is not a consequence of glassiness but due to the dependence of the activation energy of the liquid on the degree of tetrahedral ordering (see Ref. B). A similar conclusion was also derived recently (see Ref. C). I agree that there is a relationship between the maximum diffusivity and the minimum T_g , but the former may not be a direct consequence of the glassiness. It seems natural to assume that both are due to structural crossover from LDL-like to HDL-like liquids. This scenario may also explain the similar behavior observed for the mW water.

RESPONSE: We thank the reviewer for suggesting this interpretation and agree that a structural basis that explains both the maximum in diffusivity and the minimum in $T_g(P)$ is certainly

plausible. In our work, we noted a crossover from LDA-like to HDA-like structures in the same regime of pressures as the minimum in $T_g(P)$, which may relate to the arguments made in Ref. B and Ref. C. We have modified the discussion on pages 11-12 to include this idea, and included references to Refs. B and C:

“The anomalous minimum in T_g vs. P for water-like models has been shown to be associated with the locus of maximum diffusivity in the fluid as a function of pressure ($D_{max}(P)$).⁵⁸ Furthermore, supercooled water’s anomalous dynamics (of which $D_{max}(P)$ is a notable aspect) can be rationalized in terms of water’s tendency to form locally-favored structures at low temperatures and pressures, and a possible crossover from LDL-like to HDL-like liquids.^{59,60} In the present work, pressures at which T_g vs. P exhibited a negative slope corresponded to LDA-like⁶¹ structures as characterized by the $S(k)$ and the oxygen-oxygen radial distribution function ($(S(k)$ and $g(r)$, see SI Figures S2 and S3)), while a positive slope corresponded to HDA-like structures,⁶¹ and a near-zero slope corresponded to a combination of LDA-like and HDA-like local structures (SI Figure S3).⁶² This observation lends credence to the structurally-based interpretation of water’s dynamic anomalies.^{59,60}”

(7) The analysis of the nonequilibrium index X is quite interesting. The increase of dX/dT near the critical pressure clearly indicates the importance of the critical slowing down. I recommend the authors add plots of dX/dT near the onset of the X increase to Fig. 5.

RESPONSE: We agree that the dX/dT is instructive, and we have added these plots to the Supporting Information (new Figure S5) and added this comment in the main text (page 15):

“To more directly visualize these trends in the slope of X , including the potential impact of critical slowing down in TIP4P/2005 near the critical pressure, we also plot $\left|\frac{dX}{dT}\right|$ in SI Figure S5.”

(8) Recently, it has been shown that the first sharp diffraction peak in the structure factor of water may serve as a good descriptor for characterizing the LDL-like and HDL-like structures (see Ref. D; see also Ref. E on the systematic change of $S(k)$). So, plotting $S(k)$ together with $g(r)$ in Figs. S1 and S2 would be useful for readers.

RESPONSE: We have added the $S(k)$ to Figures S2 and S3 to complement the existing analysis of the $g(r)$, and also included references to the suggested Refs. D and E.

References

- [A] H. Tanaka, *Eur. Phys. J. E* **35**, 113 (2011).
- [B] R. Shi, J. Russo, H. Tanaka, *PNAS*, **115**, 9444 (2018).
- [C] R. Horstmann and M. Vogel, *J. Chem. Phys.*, **154**, 054502 (2021).
- [D] R. Shi and H. Tanaka, *JACS*, **142**, 2868 (2020).
- [E] F. Perakis, et al., *PNAS*, **114**, 8193 (2017).

REVIEWERS' COMMENTS

Reviewer #1 (Remarks to the Author):

The authors have satisfactorily answered all my questions and clarified some concept that were unclear. I am happy for the manuscript to be published.

Reviewer #2 (Remarks to the Author):

I am satisfied with the author's responses and thus recommend publication of the manuscript in its present form.

Reviewer #3 (Remarks to the Author):

I find that the authors have addressed all the comments raised by the reviewers in a satisfactory manner. This paper is an excellent work discussing the impact of the criticality of the second critical point on water's glassy state for the first time. It will provide a new promising way to detect the second critical point of water experimentally. I recommend the publication of this paper in Nature Communications as it is.